# Automated Detection and Diagnosis of Spinal Schwannomas and Meningiomas Using Deep Learning and Magnetic Resonance Imaging

**DOI:** 10.3390/jcm12155075

**Published:** 2023-08-02

**Authors:** Sadayuki Ito, Hiroaki Nakashima, Naoki Segi, Jun Ouchida, Masahiro Oda, Ippei Yamauchi, Ryotaro Oishi, Yuichi Miyairi, Kensaku Mori, Shiro Imagama

**Affiliations:** 1Department of Orthopedic Surgery, Nagoya University Graduate School of Medicine, Nagoya 466-8560, Japanmiyairi0317@yahoo.co.jp (Y.M.);; 2Information Strategy Office, Information and Communications, Nagoya University, Nagoya 464-8601, Japan; 3Department of Intelligent Systems, Nagoya University Graduate School of Informatics, Nagoya 464-8601, Japan; 4Research Center for Medical Bigdata, National Institute of Informatics, Tokyo 101-8430, Japan

**Keywords:** magnetic resonance imaging, deep learning, spinal cord neoplasms, meningioma, schwannoma

## Abstract

Spinal cord tumors are infrequently identified spinal diseases that are often difficult to diagnose even with magnetic resonance imaging (MRI) findings. To minimize the probability of overlooking these tumors and improve diagnostic accuracy, an automatic diagnostic system is needed. We aimed to develop an automated system for detecting and diagnosing spinal schwannomas and meningiomas based on deep learning using You Only Look Once (YOLO) version 4 and MRI. In this retrospective diagnostic accuracy study, the data of 50 patients with spinal schwannomas, 45 patients with meningiomas, and 100 control cases were reviewed, respectively. Sagittal T1-weighted (T1W) and T2-weighted (T2W) images were used for object detection, classification, training, and validation. The object detection and diagnosis system was developed using YOLO version 4. The accuracies of the proposed object detections based on T1W, T2W, and T1W + T2W images were 84.8%, 90.3%, and 93.8%, respectively. The accuracies of the object detection for two spine surgeons were 88.9% and 90.1%, respectively. The accuracies of the proposed diagnoses based on T1W, T2W, and T1W + T2W images were 76.4%, 83.3%, and 84.1%, respectively. The accuracies of the diagnosis for two spine surgeons were 77.4% and 76.1%, respectively. We demonstrated an accurate, automated detection and diagnosis of spinal schwannomas and meningiomas using the developed deep learning-based method based on MRI. This system could be valuable in supporting radiological diagnosis of spinal schwannomas and meningioma, with a potential of reducing the radiologist’s overall workload.

## 1. Introduction

Spinal magnetic resonance imaging (MRI) is the method of choice for diagnosing spinal ailments, including myelopathy, spinal canal stenosis, traumatic injuries, and vertebral fractures [1,2,3,4,5]. Among these, spinal cord tumors are comparatively rare. Often, these tumors remain asymptomatic and are discovered almost serendipitously [6,7,8]. Thus, their detection poses a challenge due to their potential to be easily overlooked [9]. Additionally, the manual process of detecting, diagnosing, and classifying spinal tumors is labor-intensive, particularly when dealing with multimodal MRI protocols. The diagnosis is also heavily influenced by the subjective judgments of individual radiologists [10,11]. As such, computational tools are indispensable to aid radiologists and spine surgeons in detecting and diagnosing/classifying these elusive spinal cord tumors using MRI. These tools can discern and identify subtle changes across diverse MRI settings. While deep learning, a subset of machine learning, has made notable strides in computer vision tasks [12,13,14,15], MRI data have significantly enhanced their diagnostic precision in medical imaging [16]. Nevertheless, the pragmatic application of MRI-based automated diagnosis for spinal cord tumors remains somewhat nebulous. Given the rarity of spinal cord tumors and the nuanced expertise required for their diagnosis, the adoption of automated diagnostic tools promises fewer overlooked cases, facilitates earlier diagnoses, and alleviates the pressures on specialists.

One groundbreaking deep learning strategy for object detection is the You Only Look Once (YOLO) algorithm, which has been acclaimed across varied sectors. Models based on YOLO stand out due to their real-time performance and precise object localization [16]. While YOLO’s efficacy has been affirmed across diverse applications, exploring alternate deep learning techniques for object detection remains pertinent.

For instance, in the realm of automated pavement crack detection, a study advocated for multiscale feature fusion deep neural networks to identify cracks from combined GPR B-scan and C-scan imagery [17]. Similarly, in detecting pulmonary nodules via CT scans, a study presented a 3D DCNN integrated with feature fusion and attention mechanisms [18]. These instances underscore the successful deployment of YOLO-centric object detection models in multiple disciplines.

Previously, our team unveiled an automatic detection system designed to accurately pinpoint schwannomas, leveraging deep learning via the You Only Look Once (YOLO) version 3 software paired with MRI data [19]. However, the applicability of this automated system for meningiomas, another prevalent spinal tumor, is yet to be explored. Broadening the scope of automated detection to encompass more spinal tumors could streamline radiologists’ workflows. Furthermore, while tumors are conspicuously detected on contrast-enhanced MRIs, they often elude detection on plain MRIs. Thus, enhancing the capability to spot tumors on plain MRIs holds profound clinical value.

Distinguishing between schwannomas and meningiomas prior to surgery is paramount, given their distinct surgical interventions. Due to the relatively high recurrence rate of spinal meningiomas, preventive measures, such as total durotomy or coagulation at the tumor attachment site, are advocated [20,21]. Hence, there is an urgent need for a precise, automated diagnostic system tailored for discerning between schwannomas and meningiomas.

Considering the above, this study aims to develop a novel MRI-based system using the latest version of YOLO, YOLOv4, which is a state-of-the-art deep learning-based object detection model. By leveraging the strengths of YOLO and exploring alternative methods, we aim to improve the detection, diagnosis, and classification of intradural extramedullary spinal schwannomas and meningiomas. The development of an accurate and reliable automated diagnostic system has the potential to reduce missed tumor cases, enable early detection, and alleviate the workload on specialists.

## 2. Methods

### 2.1. Patients

Between April 2015 and March 2019, the medical records of patients who underwent spinal tumor resection for schwannomas and meningiomas at the hospital were retrospectively reviewed. Schwannomas and meningiomas were diagnosed based on histological results. Ninety-five consecutive patients were included in the study (50 with schwannoma and 45 with meningioma). Other intradural extramedullary tumors, such as myxopapillary ependymomas, neurofibromas, malignant nerve sheath tumors, and metastasis were excluded (Figure 1). One hundred patients with non-spinal tumor disease (cervical myelopathy, ossification of the posterior longitudinal ligament, lumbar spinal canal stenosis, etc.) who presented to our hospital between April 2015 and March 2019 and had a spine MRI taken were selected as controls.

The participant characteristics are shown in Table 1. There were 35 male and 60 female patients (average age, 58.3 ± 14.6 years). There were 29, 47, and 19 tumors at the cervical, thoracic, and lumbar levels, respectively. There were 50 men and 50 women of the control cases, with an average age of 54.1 ± 17.6 years.

There were 24 male and 26 female patients with schwannomas (average age, 54.4 ± 15.1 years). There were 17, 15, and 18 schwannomas at the cervical, thoracic, and lumbar levels, respectively. With respect to meningiomas, there were 11 male and 34 female patients (average age, 62.8 ± 12.7 years). There were 12, 32, and 1 meningiomas at the cervical, thoracic, and lumbar levels, respectively. The proportion of women was significantly higher in the meningioma group, and lumbar occurrence was significantly higher in the schwannoma group, which is consistent with previous reports [22]. There were no significant differences in the mean age between the two groups.

### 2.2. Magnetic Resonance Imaging Dataset

All the patients with spinal tumor underwent preoperative MRI. Data on 50 MRI examinations with 122 slices for T1-weighted imaging (T1WI) and T2-weighted imaging (T2WI) were obtained from 50 patients with schwannomas. Similarly, data of 45 MRI examinations with 111 slices for T1WI and T2WI were obtained from 45 patients with meningiomas. Data of 100 MRI examinations of 200 slices for T1WI and T2WI from 100 patients with non-spinal tumor disease (cervical myelopathy, ossification of the posterior longitudinal ligament, lumbar spinal canal stenosis, etc.) were obtained as controls.

The MRI examinations were performed using either 1.5-T or 3.0-T MRI systems. The spine was examined with the following sequences and images: sagittal T2W (repetition time [TR] = 3000 ms, echo time [TE] = 90–120 ms) and sagittal T1W (TR = 400–600 ms, TE = 10–15 ms) with a 3 mm section thickness and 300 mm field of view. All the examinations were performed without an intravenous contrast agent. The MRI acquisition protocol was not universal because the hospital was equipped with 1.5-T and 3.0-T MRI scanners. Moreover, we included MRI scans acquired from other hospitals. 

Only sagittal T1W and T2W images were selected as representative images for training the object detection model. The choice was made because sagittal images encompass a broader span of spinal regions compared to axial images, making them crucial for training the object detection model.

### 2.3. Image Preparation for Deep Learning

Spinal MRI scans from Digital Imaging and Communications in Medicine files were exported using the JPEG format from the picture archiving and communication systems at our hospital. Images were annotated using a label image [23] by manually inputting a minimal bounding box containing the tumor completely and anteroposterior border of the spinal canal on sagittal MRI to generate an image for object detection training (Figure 2). Schwannoma was labeled sch and meningioma was labeled men. To ensure that the tumor was recognized, we selected several slices of the tumor (1–4 slices, depending on the case). Two slices of each control were selected. The images were trained after pixel value normalization, which replaces pixel values with real values between 0 and 1.

### 2.4. Deep Learning-Based Object Detection and Classification

Our system was developed by deep learning experts at our institute. Python (version 3.7.7; https://www.python.org, accessed on 24 June 2023), Google’s open-source Deep Learning Framework TensorFlow (version 1.14.0; https://www.tensorflow.org, accessed on 24 June 2023), and Keras (version 2.2.4; https://github.com/keras-team/keras/releases/tag/2.2.4, accessed on 24 June 2023) were used to construct the architecture for object detection and classification. The YOLO version 4 architectural model 15 was used [24]. The fourth iteration of the YOLO series, YOLOv4, was developed by Alexey Bochkovskiy, Chien-Yao Wang, and Hong-Yuan Mark Liao. This updated version is superior in speed and accuracy compared to its predecessors, and is designed for production system object detection. The detection process begins by taking an input image of size H × W. Features are extracted using networks such as VGG16, Darknet53, and ResNet50, termed as the “backbone”. This is followed by the “neck”, which draws features at different scales using networks like the Feature Pyramid Network (FPN) and Path Aggregation Network (PAN). Finally, object detection predictions are made using anchor boxes. In addition, the cutoff value of the probability of the anchor box was determined and modified so that the box with the maximum probability was detected.

The object detection model was trained using the tumor locations and labels of the schwannomas and meningiomas as the training data. The optimal probability threshold was determined by performing a few trials. Figure 3A is an example of a T2WI MRI image of a schwannoma that was tested. A region with a probability exceeding the determined threshold was detected (Figure 3B). When multiple regions were detected during trained object detection, the region exhibiting the maximum probability was chosen (Figure 3C). For classification, we labeled the schwannomas red and meningiomas blue (Figure 3C,D), and trained the object detection model using T1W and T2W images separately. Object detection and classification achieved using T1WI and T2WI could identify the region with the highest probability achieved by using T1WI and T2WI separately (Figure 4). The object detection model was trained and validated using a computer equipped with a Quadro P6000 graphics processing unit (NVIDIA, Santa Clara, CA, USA), Xeon E5-2667 v4 3.2 GHz CPU (Intel, Santa Clara, CA, USA), and 64 GB of RAM.

### 2.5. Performance Evaluation

The proposed object detection model’s performance was evaluated by five-fold cross-validation. The 433 training images (Schwannoma122, Meningioma111, no tumor200) were divided into 5 parts, one for testing (87, 87, 87, 86, 86) and one for training (346, 346, 346, 347, 347). Then, 34 images of the training data were randomly selected as validation data, and the remaining 302 or 303 were trained with data processing, making it one epoch. The performance of the training model was checked on the validation data, and the accuracy and loss function were calculated (Figure 5). The network was trained for 50 epochs. Thereafter, the accuracy of the created model was calculated using the test data.

Data augmentation helped improve the learning accuracy over the training iterations. The following augmentations were used for the collected images: image scaling—scaling from the range of quarter to twice, selected at random; left–right image flipping—flipping the image to the left–right side. Each time an image is selected, there is a 50% chance that it will be flipped left–right.

### 2.6. Image Assessment by Spine Surgeons

Two spine surgeons (NS and JO with 15 and 13 years of experience, respectively) reviewed the T1W1 and T2WI MRI images, which were identical to those used during the training of the deep learning based on object detection, and diagnosed each patient. They had no clinical information for any of the patients. This was intended to provide equal competition between object detection and the spine surgeons.

### 2.7. Statistical Analyses

All statistical analyses were performed using SPSS (version 22.0), and the results of the five-fold cross-validation of object detection were obtained. We evaluated object detection based on three criteria (true detection, false detection, and no detection) for T1WI, T2WI, and T1WI + T2WI. Data are presented as the mean ± standard deviation unless specified otherwise.

## 3. Results

Performances of the object detection and classification models are listed in Table 2. For object detection, true positive (TP), false positive (FP), false negative (FN), and true negative (TN) of the object detection using YOLO version 4 based on T1WI were 42.0%, 12.0%, 3.3%, and 42.7%, respectively. The TP, FP, FN, and TN of the object detection based on T2WI were 46.9%, 6.9%, 2.8%, and 43.4%, respectively. Moreover, the TP, FP, FN, and TN of the object detection based on T1WI + T2WI were 49.4%, 6.2%, 0%, and 44.4%, respectively. The TP, FP, FN, and TN were 45.0%, 5.8%, 5.3%, and 43.9%, for spine surgeon 1, and 46.4%, 4.2%, 5.8%, and 43.6%, respectively, for spine surgeon 2.

Regarding classification, the correct (CC) and incorrect classifications (IC) for the TP of object detection based on T1WI were 76.4% and 23.6%, respectively. The CC and IC based on T2WI were 83.3% and 16.7%, respectively. The CC and IC based on T1WI + T2WI were 84.1% and 15.9%, respectively. The CC and IC for spine surgeon 1 were 77.4% and 22.6%, respectively. The CC and IC for spine surgeon 2 were 76.1% and 23.9%, respectively.

The accuracy of location (AL = TP + TN/TP + FP + FN + TN), precision rate (TP/TP + FP), recall rate (TP/TP + FN), and accuracy of classification (AC = CC/TP) were calculated, as shown in Table 3. The AL, PR, RR, and AC of the object detection based on T1WI were 84.8%, 77.8%, 92.9%, and 76.4%, respectively. The AL, PR, RR, and AC of the object detection based on T2WI were 90.3%, 87.1%, 94.4%, and 83.3%, respectively. Moreover, the L, PR, RR, and AC of the object detection based on T1WI + T2WI were 93.8%, 88.8%, 100.0%, and 84.1%, respectively. The AL, PR, RR, and AC were 88.9%, 88.6%, 89.4%, and 77.4%, respectively, for spine surgeon 1, and 90.1%, 91.8%, 88.9%, and 76.1%, respectively, for spine surgeon 2.

### 3.1. Comparison between Schwannomas and Meningiomas

Comparison of performance between schwannomas and meningiomas is shown in Table 4. The ALs of schwannomas and meningiomas for T1WI were 77.0% and 79.3%; for T2WI, 87.7% and 86.5%; for T1WI + T2WI, 91.8% and 91.9%, respectively. There was no significant difference in ALs between the schwannomas and meningiomas.

Moreover, the ACs of schwannomas and meningiomas for T1WI were 94.7% and 56.8%; for T2WI, 94.4% and 70.8%; for T1WI + T2WI, 96.4% and 70.6%, respectively. The AC of schwannomas was higher than that of meningiomas.

### 3.2. Comparison among Tumor Levels

Comparison of performance among the cervical, thoracic, and lumbar spine is shown in Table 5. The ALs of the cervical, thoracic, and lumbar spine based on T1WI were 74.4%, 77.1%, and 86.0%; 84.6%, 89.5%, and 86.0% based on T2WI; and 92.3%, 86.0%, and 94.0% based on T1WI + T2WI, respectively. There was no significant difference in ALs between the cervical, thoracic, and lumbar spine. 

The ACs of the cervical, thoracic, and lumbar spine based on T1WI were 69.0%, 71.6%, and 95.3%; 77.3%, 81.9%, and 95.3% based on T2WI; and 75.0%, 84.2%, and 97.9% based on T1WI + T2WI, respectively. The AC of the lumbar spine was higher than the other regions.

### 3.3. External Data Validation

Performance was evaluated using 60 additional data (15 schwannomas, 15 meningiomas, and 30 controls) that were not used during this model construction. The characteristics of the additional data are shown in Table 6. One slice from each patient was used for validation. Performances of the object detection and classification models are listed in Table 7. The AL, PR, RR, and AC of the object detection based on T1WI were 86.7%, 83.3%, 89.3%, and 76.0%, respectively. The AL, PR, RR, and AC of the object detection based on T2WI were 91.7%, 90.0%, 93.1%, and 81.5%, respectively. Moreover, the AL, PR, RR, and AC of the object detection based on T1WI + T2WI were 93.3%, 90.3%, 96.6%, and 82.1%, respectively.

## 4. Discussion

The performance of our object detection model was 93.8%, compared to the spine surgeons’ performances of 88.9% and 90.1%. Our model’s classification accuracy was 84.1%, while the surgeons’ performances were 77.4% and 76.1%. Our model surpassed the surgeons in both detection and classification accuracy. The system can sift through all sagittal T1WI and T2WI scans, capturing and diagnosing incidental spinal schwannomas and meningiomas. This capability aims to enhance radiographical diagnoses, reduce oversight of rare spinal tumors, and assist in tumor type identification.

Our system’s ability to distinguish between schwannomas and meningiomas mirrors the performance reported by radiological specialists using plain and contrast-enhanced MRI [25,26]. Moreover, our results were superior to the participating spine surgeons diagnosing the same conditions. Prior studies have detailed the distinct characteristics of schwannomas and meningiomas [27,28,29]. However, certain clinical cases can pose challenges in distinguishing tumors based on intensity, location, configuration, and shape [4,27,28]. Hence, MRI-based differentiation is not always dependable [27,29,30].

There are several reports on the automated interpretation of radiological findings of spinal cord tumors. However, most of these systems are limited in their capabilities, often only detecting the location of the spinal cord tumor or classifying them. For instance, one prior study discussed automated object detection specific to spinal schwannomas. Additionally, Maki et al. presented the automated classification of schwannomas and meningiomas using plain and enhanced MRI [19,25]. Our research, therefore, marks the first effort to develop an automated diagnostic system capable of both “detecting” and “classifying” spinal schwannomas and meningiomas with high precision using only a single slice of a plain MRI sagittal image (T1 and T2WI). Notably, our system solely analyzes single-sliced images. Given the inherent challenges specialists face when distinguishing between schwannomas and meningiomas using only single-sliced images, this is a significant advancement. Maki et al.’s study on spinal tumor classification integrated multiple (axial, sagittal, and coronal) images from plain and contrast-enhanced MRI and achieved an accuracy of 81% [25]. Remarkably, our system, requiring only one MRI sagittal image for its analysis, boasts a comparable accuracy rate of 84.1%, even with limited data. This system offers numerous advantages, including its rapid transfer and analysis times, making it highly efficient for remote and early diagnoses. Moreover, its ability to identify tumors appearing in just one MRI slice enhances its capability to detect smaller lesions.

Regarding the differentiation between schwannomas and meningiomas, our system can classify lesions as schwannomas with impressive accuracy. However, occasionally, meningiomas were misclassified as schwannomas, a trend also noted in prior automated classification research [25]. We chose not to incorporate contrast-enhanced MRI in our current system, even though an enhanced MRI often aids in diagnosing spinal tumors [31]. This is because some patients might be allergic to the contrast medium or suffer from renal dysfunction, among other issues. For these individuals, our system is especially beneficial, as it can accurately detect and classify schwannomas and meningiomas using only plain MRI. Additionally, it assists in determining if further imaging tests, such as myelography CT or enhanced MRI, are warranted.

In our study, the system assessed all spinal regions, from cervical to lumbar, consistently detecting the location with great accuracy across all levels. A challenge, however, was the variability in MRI systems and acquisition settings utilized in our research, leading to a range in MRI quality. Nevertheless, deep learning using images under diverse conditions might empower convolutional neural networks to better recognize novel images [32]. We believe that incorporating a variety of MRI systems and acquisition settings could enhance the system’s image recognition capabilities. Given its adaptability, our system can be deployed across different facilities, delivering precise diagnoses across various spinal regions and MRI configurations.

Though our study had a limited number of cases, the system demonstrated impressive location detection and classification using just one image. We augmented our training data using several enhancements. This allowed us to maximize results from the available data by selecting the optimal epoch and ensuring adequate training [33]. Going forward, research should expand to cover a larger number of cases, especially rare spinal cord tumors beyond schwannomas and meningiomas, as this may further improve accuracy rates.

This study has several limitations. First, the number of MRI images utilized was relatively limited, indicating the need to enhance our system’s accuracy using more MRI images. Nonetheless, through data augmentation of the available MRI images, the proposed system achieved a performance in line with prior reports [34]. Data augmentation bolsters training datasets by introducing random transformations, such as flipping and scaling. This approach proves beneficial for deep learning, especially when working with limited datasets. Although the current system focuses solely on detecting and classifying spinal schwannomas and meningiomas, the inclusion of other spinal tumors is essential for broader clinical applicability in future research. However, given that spinal schwannomas and meningiomas are among the most prevalent spinal tumors [22], we assert this study’s clinical significance. We have not yet validated our system against other spinal conditions like disc herniation or Tarlov cyst, leaving the potential for false positives open. Nonetheless, our system has reliably identified the absence of tumors in non-tumor cases, underlining its efficacy in pinpointing abnormalities. For this research, we focused on individual slices of the MRI’s sagittal plane. This facilitated detection along the midline, distinguishing between nerve roots and diminutive tumors on the more lateral plane proved challenging. Future iterations should prioritize enhanced accuracy by incorporating learning from standard lateral slices. Moreover, our sole reliance on sagittal images means incorporating axial and coronal images could further optimize the system’s performance. Employing DICOM data might also boost accuracy, given that our current jpeg images inherently possess lower information content. However, the streamlined nature of our system, emphasizing sagittal jpeg images, ensures quick analyses. This simplicity and efficiency suggest the system’s suitability across diverse MRI configurations at various institutions.

## 5. Conclusions

In this research, we introduced a groundbreaking object detection and classification system tailored for spinal schwannomas and meningiomas using MRI data. The system demonstrated impressive performance, especially when juxtaposed against the evaluations of seasoned spine surgeons. With the ability to detect incidental schwannomas and meningiomas on plain MRI, our system offers a promising avenue for effective screening. Such an innovation has the potential to significantly mitigate the workload of radiologists.

Nevertheless, it is crucial to highlight the limitations of our study. The number of MRI images employed was relatively modest, which underlines the need for more extensive datasets to refine our system’s accuracy further. While our approach predominantly targeted spinal schwannomas and meningiomas, there is a vast spectrum of other spinal tumors that remain unexplored, a direction that future research could pursue. The potential interference of other spinal diseases such as disc herniation or Tarlov cyst and their possible identification as false positives is an aspect we aim to address in future iterations of the system. Moreover, the present study mainly engaged with sagittal images; incorporating axial and coronal images could substantially enhance the system’s efficacy.

Data augmentation played a pivotal role in optimizing the performance of our system. By leveraging techniques such as random transformations, flipping, and scaling, we were able to enrich our training datasets despite their initial limitations. Such approaches underscore the importance of adaptability in deep learning, especially when working with constrained datasets.

In summation, while our study marks a significant stride in the domain of automated tumor detection and classification, there is a vast expanse of possibilities that lie ahead. Continuous refinement, broader dataset incorporation, and exploration of other tumor types can propel our system to newer horizons, ultimately reshaping the landscape of radiological diagnosis.

## Figures and Tables

**Figure 1 jcm-12-05075-f001:**
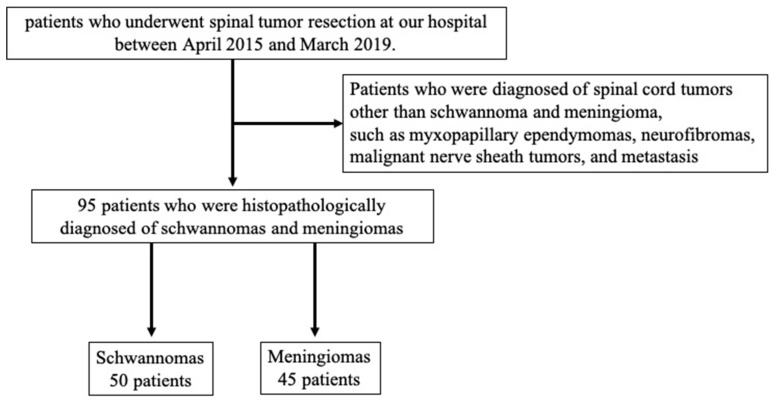
Flowchart of patient selection.

**Figure 2 jcm-12-05075-f002:**
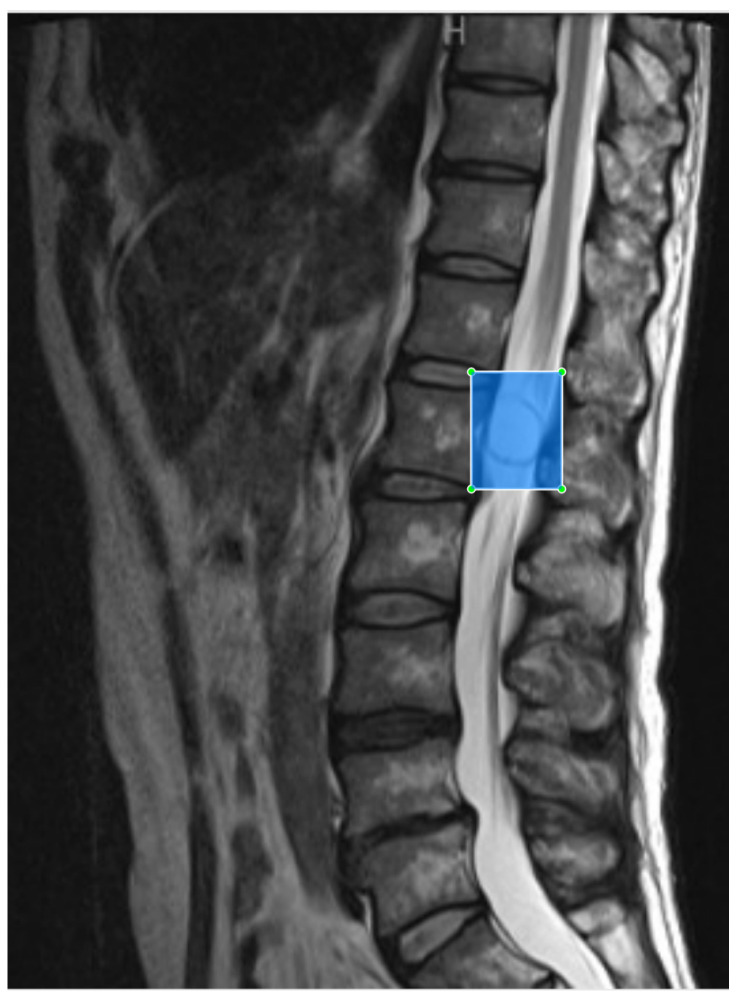
Preparation of images for training the object detection model. Tumor recognition slices were selected and placed within a minimal bounding box on the sagittal plane during magnetic resonance imaging.

**Figure 3 jcm-12-05075-f003:**
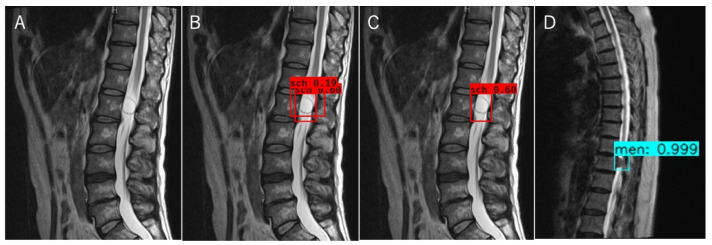
Object detection and classification method. (**A**–**C**) Process of object detection for the T2WI MRI image of a schwannoma; (**A**) T2WI; (**B**) The regions above the probability threshold were detected by the trained object detection model using T2WI; (**C**) The final region with the highest probability shown in (**B**) was selected and classified as a schwannoma; (**D**) The result of object detection for the T2WI MRI image of a meningioma, T2WI, T2-weighted imaging.

**Figure 4 jcm-12-05075-f004:**
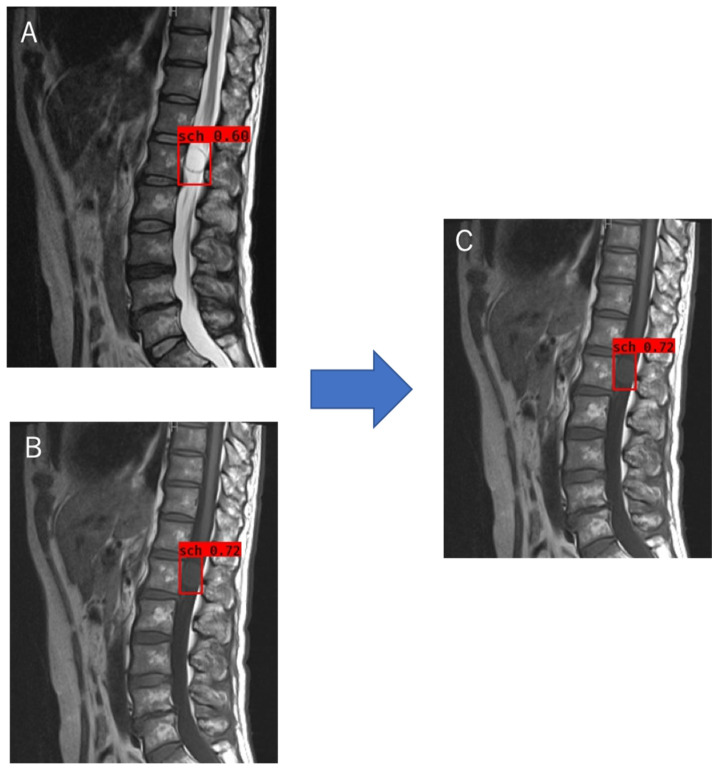
Object detection and classification method using T1WI and T2WI. (**A**) The final region in T2W images; (**B**) The final region in T1W images; (**C**) The region with a higher probability shown in (**A**,**B**) was selected as the final region for T2WI and T1WI. T1WI, T1-weighted imaging; T2WI, T2-weighted imaging.

**Figure 5 jcm-12-05075-f005:**
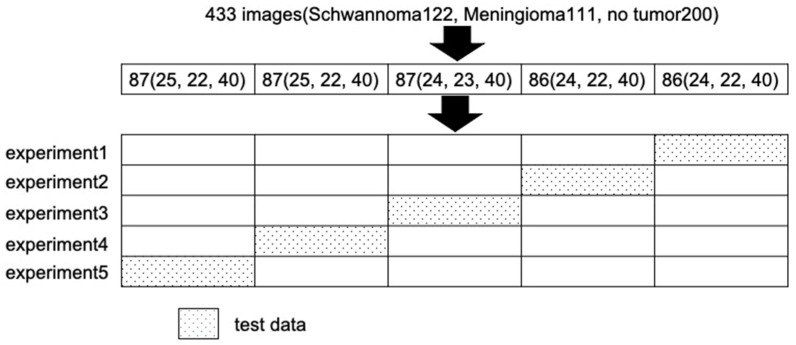
Training process. The 433 training images were divided into five parts, one for testing (87, 87, 87, 86, 86) and one for training (346, 346, 346, 347, 347). Each group ran its own learning. The accuracy of the created model was calculated using the test data.

**Table 1 jcm-12-05075-t001:** Baseline characteristics of the patients.

		Total	Schwannoma	Meningioma	Controls
Number of patients		95	50	45	100
Sex (male/female)		35/60	24/26	11/34	50/50
Age (years)		58.3 ± 14.6	54.4 ± 15.1	62.8 ± 12.7	55.2 ± 16.6
Height (cm)		159.4 ± 10.2	162.2 ± 10.5	156.3 ± 8.9	160.1 ± 11.1
Weight (kg)		59.1 ± 14.2	62.3 ± 12.8	55.3 ± 14.9	60.8 ± 15.2
Level of the tumor	Cervical	29	17	12	n.a.
	Thoracic	47	15	32	n.a.
	Lumbar	19	18	1	n.a.

Values are presented as the mean ± standard deviation for each group; n.a., not applicable.

**Table 2 jcm-12-05075-t002:** Object detection and classification performance of our system.

	MRI	Detection (%)				Classification (%)	
		TP	FP	FN	TN	CC	IC
Object detection	T1	42	12	3.3	42.7	76.4	23.6
	T2	46.9	6.9	2.8	43.4	83.3	16.7
	T1 + T2	49.4	6.2	0	44.4	84.1	15.9
Spine surgeon	1	45.0	5.8	5.3	43.9	77.4	22.6
	2	46.4	4.2	5.8	43.6	76.1	23.9

TP, true positive; FP, false positive; FN, false negative; TN, true negative; CC, correct classification; IC, incorrect classification; MRI, magnetic resonance imaging.

**Table 3 jcm-12-05075-t003:** Object detection and classification accuracy of our system.

	MRI	AL (%)	PR (%)	RR (%)	AC (%)
Object detection	T1	84.8	77.8	92.9	76.4
	T2	90.3	87.1	94.4	83.3
	T1 + T2	93.8	88.8	100.0	84.1
Spine surgeon	1	88.9	88.6	89.4	77.4
	2	90.1	91.8	88.9	76.1

Accuracy of location (AL) = TP/(TP + FP + FN). Precision rate (PR) = TP/(TP + FP). Recall rate (RR) = TP/(TP + FN). Accuracy of classification (AC) = CC/(CC + IC). TP, true positive; FP, false positive; FN, false negative; CC, correct classification; IC, incorrect classification; MRI, magnetic resonance imaging.

**Table 4 jcm-12-05075-t004:** Object detection and classification accuracy of our system for schwannomas and meningiomas.

	MRI	AL (%)		AC (%)	
		Schwannoma	Meningioma	Schwannoma	Meningioma
Object detection	T1	77.0	79.3	94.7	56.8
	T2	87.7	86.5	94.4	70.8
	T1 + T2	91.8	91.9	96.4	70.6

Accuracy of location (AL) = TP/(TP + FP + FN). Accuracy of classification (AC) = CC/(CC + IC). TP, true positive; FP, false positive; FN, false negative; CC, correct classification; IC, incorrect classification; MRI, magnetic resonance imaging.

**Table 5 jcm-12-05075-t005:** Object detection and classification accuracy of our system for the cervical, thoracic, and lumbar spine.

	MRI	AL (%)			AC (%)		
		Cervical	Thoracic	Lumbar	Cervical	Thoracic	Lumbar
Object detection	T1	74.4	77.1	86.0	69.0	71.6	95.3
	T2	84.6	89.5	86.0	77.3	81.9	95.3
	T1 + T2	92.3	86.0	94.0	75.0	84.2	97.9

Accuracy of location (AL) = TP/(TP + FP + FN). Accuracy of classification (AC) = CC/(CC + IC). TP, true positive; FP, false positive; FN, false negative; CC, correct classification; IC, incorrect classification; MRI, magnetic resonance imaging.

**Table 6 jcm-12-05075-t006:** Baseline characteristics of the patients for external validation.

		Total	Schwannoma	Meningioma	Controls
Number of patients		30	15	15	30
Sex (male/female)		11/19	7/8	4/11	15/15
Age (years)		59.2 ± 11.9	58.7 ± 12.8	59.9 ± 11.3	54.8 ± 18.6
Height (cm)		157.9 ± 8.2	159.7 ± 7.5	155.8 ± 8.7	159.8 ± 13.1
Weight (kg)		54.9 ± 9.8	56.7 ± 12.3	53.0 ± 5.9	60.1 ± 16.2
Level of the tumor	Cervical	11	6	5	n.a.
	Thoracic	13	4	9	n.a.
	Lumbar	6	5	1	n.a.

Values are presented as the mean ± standard deviation for each group; n.a., not applicable.

**Table 7 jcm-12-05075-t007:** Performance of object detection and classification accuracy of our system using external data.

	MRI	AL (%)	PR (%)	RR (%)	AC (%)
Object detection	T1	86.7	83.3	89.3	76.0
	T2	91.7	90.0	93.1	81.5
	T1 + T2	93.3	90.3	96.6	82.1

Accuracy of location (AL) = TP/(TP + FP + FN). Precision rate (PR) = TP/(TP + FP). Recall rate (RR) = TP/(TP + FN). Accuracy of classification (AC) = CC/(CC + IC). TP, true positive; FP, false positive; FN, false negative; CC, correct classification; IC, incorrect classification; MRI, magnetic resonance imaging.

## Data Availability

The data of health checkups used to support the findings of this study are available from the corresponding author upon request.

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
