# Peer review of "Automated Detection and Diagnosis of Spinal Schwannomas and Meningiomas Using Deep Learning and Magnetic Resonance Imaging"

_jcm, 2023, doi:10.3390/jcm12155075_

Round 1

Reviewer 1 Report (New Reviewer)

The authors have proposed a deep-learning method for automatic classification of spinal tumors. While the research question is timely and relevant the technical implementation is insufficient to justify publication in JCM at this point. Mainly because the authors used internal validation, i.e. the algorithm is continuously trained on the same dataset. These days, it is absolutely necessary to also show external validation on a novel test set. This shortcoming is unfortunately also not discussed in the limitation section.

Secondly, the authors used a workflow of DICOM-->JPG export and trained the network on 2D slices, thereby reducing the potential data available for the training of the network significantly. While this shortcoming is discussed, I believe the algorithms performance can be significantly increased if the 3D DICOM datasets are directly used to train the algorithm.

Fine.

Author Response

Reviewer 2 Report (New Reviewer)

Moderate editing of English language required

Round 2

Reviewer 1 Report (New Reviewer)

The authors have addressed my concerns. Thank you.

Some language editing is still required, e.g. "Lumber" --> lumbar!

Author Response

Reviewer 2 Report (New Reviewer)

Accept for publication

Minor editing of English language required

Author Response

This manuscript is a resubmission of an earlier submission. The following is a list of the peer review reports and author responses from that submission.

Round 1

Reviewer 1 Report

This is an interesting paper. I enjoy reading it. The authors developed an AI software to detect the spinal schwannomas and meningiomas. Interestingly, the AI was able to discriminate the meningiomas from schwannomas. I think the authors did a good job. I have only some minor comments for the authors:

- The authors should emphasize the potential clinical implementation of this software in screening setting, especially for patients coming for unspecific low back pain. Because these tumors can be easily missed on routine MRI workup without contrast administration. In this senario, detection accuracy is much more important than discriminating between these tumors. Because in reality, we need more than just plain MRI to discriminate these two tumors. For example, intravenous contrast administration.

- The authors didn't test the detecting ability of the software in other patients cohort. I wonder how the software perform in cases of disc herniation, or Tarlov cyst? Or in cases of spondyloarthritis with spinal stenosis...? I doubt that the accuracy would be compromised.

- In conclusion, the authors stated that "The proposed system’s accuracy was equal to or higher than that reported by radiologists who used simple and contrast-enhanced MRI for diagnosis." This statement is unclear, and I think, is incorrect. In this study, you didn't compare the software with human. So you cannot say that. Furthermore, an experienced radiologist can hardly miss a spinal schwannoma or meningioma with gadolinium enhanced MRI, because most of these tumors will show strong contrast enhancement. The role of AI is, again, in screening, before we give i.v. contrast. 

Additional comments

- Please delete "This study was approved by our institutional review board, and the requirement for consent was waived considering the retrospective nature of the analyses" at line 69-70. It was already mentioned at line 66-67.

- You stated in line 318-319 that "Informed Consent Statement: Informed consent was obtained from all subjects involved in the study." This doesn't match with the statement above that " requirement for consent was waived considering the retrospective nature of the analyses"

Author Response

This is an interesting paper. I enjoy reading it. The authors developed an AI software to detect the spinal schwannomas and meningiomas. Interestingly, the AI was able to discriminate the meningiomas from schwannomas. I think the authors did a good job. I have only some minor comments for the authors:

- The authors should emphasize the potential clinical implementation of this software in screening setting, especially for patients coming for unspecific low back pain. Because these tumors can be easily missed on routine MRI workup without contrast administration. In this senario, detection accuracy is much more important than discriminating between these tumors. Because in reality, we need more than just plain MRI to discriminate these two tumors. For example, intravenous contrast administration.

Thank you for pointing this out.

We think the most clinically useful part of this study is to avoid missing incidentally present tumors.

We added the following.

Page2 line54

In addition, a plain MRI is usually performed for close examination when a patient presents with nonspecific neck or back pain. Tumors are easily detected on contrast-enhanced MRI, but are often difficult to detect on plain MRI, so the ability to detect tumors on plain MRI is of great clinical significance.

- The authors didn't test the detecting ability of the software in other patients cohort. I wonder how the software perform in cases of disc herniation, or Tarlov cyst? Or in cases of spondyloarthritis with spinal stenosis...? I doubt that the accuracy would be compromised.

Thank you for pointing this out.
Since we have not validated this system with disc herniation or Tarlov cyst, we cannot rule out the possibility of them being false positives. We will add this to the limitation. However, we have been able to determine the absence of tumor with sufficient accuracy in cases with no tumor, and we believe that this system is significant in terms of efficient detection of abnormal findings.

Page9 line296

Since we have not validated this system with other spinal diseases such as disc herniation or Tarlov cyst, we cannot rule out the possibility of them being false positives. However, we have been able to determine the absence of tumor with sufficient accuracy in cases with no tumor, and we believe that this system is significant in terms of efficient detection of abnormal findings.

- In conclusion, the authors stated that "The proposed system’s accuracy was equal to or higher than that reported by radiologists who used simple and contrast-enhanced MRI for diagnosis." This statement is unclear, and I think, is incorrect. In this study, you didn't compare the software with human. So you cannot say that. Furthermore, an experienced radiologist can hardly miss a spinal schwannoma or meningioma with gadolinium enhanced MRI, because most of these tumors will show strong contrast enhancement. The role of AI is, again, in screening, before we give i.v. contrast. 

As you indicated, no comparison was made between this system and radiologists in the cases used in this study. We have compared them to previous reports and found them to be comparable. As you point out, if contrast is performed to look for tumors, it is almost impossible to miss them.
The part of this study that is of high clinical significance is the incidental detection without contrast and unrelated to symptoms.
The conclusion has been changed as follows.

Page9 line308

5. Conclusion

We developed a novel object detection and classification system for spinal schwannomas and meningiomas that used MRI data. The proposed system can detect incidental schwannomas and meningiomas on plain MRI and is useful as a screening tool. Thus, this system may reduce the overall workload for radiologists.

Additional comments

- Please delete "This study was approved by our institutional review board, and the requirement for consent was waived considering the retrospective nature of the analyses" at line 69-70. It was already mentioned at line 66-67.

Thank you for pointing that out.

I have removed it.

- You stated in line 318-319 that "Informed Consent Statement: Informed consent was obtained from all subjects involved in the study." This doesn't match with the statement above that " requirement for consent was waived considering the retrospective nature of the analyses"

Thank you for pointing this out.

We have made the following correction.

Page10 line328

The requirement for informed consent was waived considering the retrospective nature of the analyses.

Reviewer 2 Report

The authors use deep learning methods on spinal MRI images to detect and classify schwannomas and meningiomas. I have several issues with this manuscript:

-) The authors argue that spinal tumors are rare and often overlooked. However, the only citation the authors back up their claime (#9) refers rather to poor clinical examination or MRI scans at the incorrect level. Once you perform a MRI scan of the correct level, spinal tumors are easily detectable.

-) It would be interesting to distinguish other spinal tumors rather than meningioma and schwannoma. e.g. myxopapillary ependymoma, ependymomas, astrocytomas, and so on. Because the authors only focus on meningiomas and schwannomas, the novelty and clinical significance of the paper remain low.

-) Additionally, the object detection rate is low (see table 3). However, a comparison with at least two human radiologists would be interesting.

-) At least, the authors could present some images from selected cases to see where the deep learning algorithm works and where it struggles.

Author Response

The authors use deep learning methods on spinal MRI images to detect and classify schwannomas and meningiomas. I have several issues with this manuscript:

-) The authors argue that spinal tumors are rare and often overlooked. However, the only citation the authors back up their claime (#9) refers rather to poor clinical examination or MRI scans at the incorrect level. Once you perform a MRI scan of the correct level, spinal tumors are easily detectable.

If the tumor is of symptomatic size, as you have pointed out, it can be identified by MRI imaging with the tumor in mind. The clinical significance of this study is to reduce the risk of missing tumors that are incidentally found on MRI scans taken for back pain and other conditions.

The conclusion has been changed as follows.

Page0 line308

5. Conclusion

We developed a novel object detection and classification system for spinal schwannomas and meningiomas that used MRI data. The proposed system can detect incidental schwannomas and meningiomas on plain MRI and is useful as a screening tool. Thus, this system may reduce the overall workload for radiologists.

-) It would be interesting to distinguish other spinal tumors rather than meningioma and schwannoma. e.g. myxopapillary ependymoma, ependymomas, astrocytomas, and so on. Because the authors only focus on meningiomas and schwannomas, the novelty and clinical significance of the paper remain low.

You are correct. We plan to accumulate cases and detect other tumors in the future. We believe that schwannoma and meningioma are more significant than other tumors as the first tumors to be verified because they are the most common tumors and have a higher probability of being found than other tumors.

We added the following to the limitation.

Page9 line292

As the proposed system only detected and classified spinal schwannomas and meningiomas, other spinal tumors will need to be investigated in the future in order to gain clinical significance. However, we believe that this study is clinically significant because spinal schwannomas and meningiomas are the most common spinal tumors. [31] Since we have not validated this system with other spinal diseases such as disc herniation or Tarlov cyst, we cannot rule out the possibility of them being false positives. However, we have been able to determine the absence of tumor with sufficient accuracy in cases with no tumor, and we believe that this system is significant in terms of efficient detection of abnormal findings.

-) Additionally, the object detection rate is low (see table 3). However, a comparison with at least two human radiologists would be interesting.

Thank you for your valuable comments.
In this study, we examined each slice in the sagittal plane of MRI.
Therefore, we believe that the detection rate is not low because it is difficult for a radiologist or spine surgeon to find a tumor in only one slice in the lateral slice. If all slices were included, the results would exceed the accuracy of the present study. In this study, it is important not to miss accidental tumors, and to enable the detection of even small tumors that are located cephalad, caudal, or lateral in the image, we examined each slice of the MRI sagittal plane and calculated the accuracy of each slice.
We believe that comparisons with radiologists are very important. We have not been able to do so in this study, so we are unable to present the results.

-) At least, the authors could present some images from selected cases to see where the deep learning algorithm works and where it struggles.

Thank you for your valuable comments.

In this study, we examined each slice in the sagittal plane of the MRI. Therefore, while detection was easy in the midline, it was difficult to discriminate between nerve roots and small tumors in the more lateral plane. In the future, the accuracy should be increased by learning normal lateral slices

This is a note of caution for this system and was added to the limitation as follows.

Page9 line300

In this study, we examined each slice in the sagittal plane of the MRI. Therefore, while detection was easy in the midline, it was difficult to discriminate between nerve roots and small tumors in the more lateral plane. In the future, the accuracy should be increased by learning normal lateral slices.

Round 2

Reviewer 2 Report

The authors have addressed a few points, however, several main issues remain in this manuscript including that the authors only integrated schwannomas and meningiomas in their analysis and no comparison with a human radiologists were conducted.